# Multiplex PCR assay to identify clinically important *Aeromonas* species

Aki Sakurai,[1,2] Naoto Hosokawa,[3] Daisuke Ohkushi,[4] Sohei Harada,[5] Yasufumi Matsumura,[6] Naoya Itoh,[7] Kazuhiro Ishikawa,[8] Sho Saito,[2] Takayuki Sakurai,[9] Ryota Hase,[10] Takehiro Hashimoto,[11] Yohei Doi,[1,12,13,14] Masahiro Suzuki[12,13]

**ABSTRACT**  The genus *Aeromonas* is increasingly implicated in human infections. However, accurate species-level identification remains challenging, particularly in clinical microbiology laboratories. This study aimed to develop a multiplex polymerase chain reaction (PCR) assay to identify four *Aeromonas* species—*Aeromonas hydrophila*, *Aeromonas caviae*, *Aeromonas veronii*, and *Aeromonas dhakensis*—most frequently associated with human infectious diseases. A total of 788 whole genome sequencing (WGS) data sets from 31 *Aeromonas* species were analyzed to identify open reading frames (ORFs) specifically present in *A. hydrophila*, *A. caviae*, *A. veronii*, and *A. dhakensis*. Primer sets were designed based on sequences of ORFs specific to each species to develop a multiplex PCR assay. To validate the efficacy of the assay, 256 clinical *Aeromonas* isolates were tested, and the results were compared with taxonomic affiliation inferred by WGS data, along with 19 type strains. The multiplex PCR successfully identified all strains of the four target species and produced no amplification in non-target species strains except the band for internal control. The multiplex PCR enables rapid and reliable identification of four *Aeromonas* spp. commonly involved in human infectious diseases.

**IMPORTANCE**  The multiplex PCR assay facilitates accurate identification of clinically important *Aeromonas* spp. in clinical microbiology laboratories, providing crucial information to guide appropriate antimicrobial therapy and advance understanding of the epidemiology of *Aeromonas* spp.

**KEYWORDS**  *Aeromonas*, multiplex polymerase chain reaction, open reading frame, species identification, rapid identification technique, human infections

**Peer Reviewer** Michael Wehrhahn, Douglass Hanly Moir Pathology, Macquarie Park, New South Wales, Australia

Address correspondence to Yohei Doi, yoheidoi@fujita-hu.ac.jp.

N.H. reports speaker payments from Meiji Seika Pharma, GSK, MSD, and Gilead. S.H. reports speaker payments from MSD, Shionogi, and Pfizer and consulting fees from Shionogi, Pfizer, and Denka outside the submitted work. N.I. reports speaker payments from Shionogi, Pfizer, MSD, Meiji Seika Pharma, BD, GSK, Asahi Kasei Pharma, bioMérieux Japan Ltd, AstraZeneca, Gilead, and Shimadzu Co. Ltd. outside the submitted work. S.S. has received a grant from Shionogi. Y.D. reports speaker payment from BD, Shionogi, and Gilead and consulting fees from Moderna, Pfizer, GSK, AbbVie, Shionogi, and Meiji Seika Pharma outside the submitted work. M.S. has received a grant from KANTO Chemical Co. The other authors declare that they have no conflict of interest.

See the funding table on p. 8.

The genus *Aeromonas* belongs to the family *Aeromonadaceae* and comprises Gram-negative, facultative anaerobic bacilli ubiquitous in aquatic ecosystems (1). *Aeromonas* spp. have been increasingly recognized as important pathogenic microorganisms not only to fish and other poikilothermic animals but also to humans (2, 3). To date, they have been linked to various human infectious diseases, including skin and soft tissue infections, bloodstream infections, gastroenteritis, and hepatobiliary infections in both immunocompetent and immunocompromised hosts (4).

Of the more than 30 species in the genus *Aeromonas*, four—*Aeromonas hydrophila*, *Aeromonas caviae*, *Aeromonas veronii*, and *Aeromonas dhakensis*—are strongly linked to human infectious diseases, collectively accounting for over 90%–98% of all cases (2, 4, 5). Interestingly, these *Aeromonas* spp. exhibit unique, species-specific antimicrobial resistance profiles due to chromosomally-encoded β-lactamase genes, including $bla_{CphA}$, which encodes class B metallo-β-lactamases; $bla_{AmpC}$, which encodes class C cephalosporinases; and $bla_{OXA}$, which encodes class D oxacillinases (5, 6). More specifically, $bla_{CphA}$ is present in *A. hydrophila*, *A. dhakensis*, and *A. veronii*, while $bla_{AmpC}$ is harbored

by *A. hydrophila* ($bla_{CpeH/CeS}$), *A. dhakensis* ($bla_{AQU}$), and *A. caviae* ($bla_{MOX}$). These genes are highly preserved in unique combinations within each species, with prevalence rates ranging from 92% to 100% (5). Consequently, effective antimicrobial agents vary between species. For example, carbapenems might not be an ideal therapeutic option for serious infections caused by *A. hydrophila*, *A. dhakensis*, and *A. veronii* due to the potential risk of treatment failure, as reported in clinical cases (7–9), while this concern does not apply to infections caused by wild-type *A. caviae* strains. Nevertheless, the presence of intrinsic resistance is often difficult to predict through phenotypic testing due to the induced expression of these enzymes following exposure to β-lactam agents (10–12). Given the unique antimicrobial resistance profile of each *Aeromonas* spp., accurate speciation is thus essential for selecting appropriate antimicrobial agents, particularly in critically ill cases. However, neither conventional biochemical tests nor automated systems reliably differentiate clinically relevant *Aeromonas* spp. (3, 13). While matrix-assisted laser desorption/ionization mass spectrometry time of flight (MALDI-TOF MS) holds promise, it still lacks sufficient discriminatory power to identify clinical *Aeromonas* spp. at the species level (13, 14). This limitation is partly due to an incomplete database, which lacks certain species and, in some cases, contains too few reference spectra, hindering proper identification (15–17).

Accurate species identification is also essential for understanding clinical features, epidemiology, and pathogenicity of *Aeromonas* infection. Recent studies have further highlighted its clinical significance in human health (18, 19). Widely distributed in the aquatic environment, *Aeromonas* can be acquired through direct mucocutaneous contact with various sources or by ingesting contaminated food or water (2). However, much remains to be understood regarding regional differences in species distribution as well as the infectivity and virulence of each species in discrete organs, such as the skin and the gastrointestinal tract (18, 20, 21).

This study aimed to develop a one-step multiplex polymerase chain reaction (PCR) assay for the rapid identification of four *Aeromonas* spp. commonly encountered in clinical settings. Open reading frame (ORF) sequences specific to target species were identified from publicly available whole genome sequencing (WGS) data, and primers were designed based on these sequences. The assay's performance was validated using clinical *Aeromonas* strains and type/reference strains from several genera.

## MATERIALS AND METHODS

### Bacterial strains and species identification

A total of 256 clinical *Aeromonas* strains were used. These strains were collected from a previously reported observational cohort study conducted in Japan between June 2020 and August 2022 (5), with the addition of clinical isolates collected at five hospitals in Japan between June 2016 and December 2023. These strains were isolated from various sources, with bile being the most common (33%, 85/256), followed by blood (32%, 83/256), stool (8.2%, 21/256), sputum (3.9%, 10/256), wound (3.1%, 8/256), and intra-abdominal abscess (3.1%, 8/256). The taxonomic affiliation of clinical *Aeromonas* strains was determined based on WGS data using orthologous average nucleotide identity (ANI), with a cut-off value of 95% used for species delineation, as reported in previous studies (5, 22, 23). For strains with WGS data available from the previous study, the existing data were used. For those without prior WGS data, sequencing was performed as described below. Reference genomes representing 31 *Aeromonas* spp. used for ANI calculation are listed in Table S1. Additionally, one *Aeromonas* American Type Culture Collection (ATCC) type strain (*A. hydrophila* ATCC 7966[T]) and 10 Japan Collection of Microorganisms Japan Collection of Microorganisms (JCM) type strains—*Aeromonas allosaccharophila* JCM 8576[T], *A. caviae* JCM 1043[T], *Aeromonas enteropelogenes* JCM 8355[T], *Aeromonas eucrenophila* JCM 8238[T], *Aeromonas jandaei* JCM 8316[T], *Aeromonas media* JCM 2385[T], *Aeromonas schubertii* JCM 7373[T], *Aeromonas sobria* JCM 2139[T], *A. veronii* JCM 7375[T], *Aeromonas salmonicida* subsp. *salmonicida* JCM 7874 [T]—were included.

Furthermore, eight non-*Aeromonas* ATCC or JCM strains (*Escherichia coli* ATCC 25922, *Klebsiella pneumoniae* ATCC 700603, *Enterobacter cloacae* subsp. *cloacae* ATCC 13047[T], *Pseudomonas aeruginosa* ATCC 27853, *Vibrio fluvialis* JCM 1281[T], *Vibrio parahaemolyticus* JCM 32818[T], *Vibrio alginolyticus* JCM 32963, and *Grimontia hollisae* JCM 1283[T]) were used to assess the specificity of the multiplex PCR.

## Whole genome sequencing

Genomic DNA was extracted from clinical isolates using the DNeasy Blood and Tissue Kit (Qiagen, Tokyo, Japan). The extracted DNA was used to prepare a sequencing library with the QIAseq FX DNA Library Kit (Qiagen, Tokyo, Japan). Whole genome sequencing was performed with the NextSeq 2000 platform (Illumina, Inc., San Diego, CA, USA) using 2 × 150 bp paired-end reads, followed by *de novo* genome assembly with SPAdes v3.13.1 (24).

## Preparation of template DNA for PCR

Bacterial colonies were suspended to a McFarland turbidity of 1 in 100 µL Tris-EDTA buffer (pH 8.0). The suspension was then heated at 99°C for 10 min and centrifuged at 13,000 rpm for 1 min. The supernatants were used as DNA templates for PCR.

## Screening of species-specific ORFs and designing of primers

Candidate ORFs serving as species-specific markers for *A. hydrophila*, *A. caviae*, *A. veronii,* and *A. dhakensis* were identified by a previously reported approach (25, 26). First, WGS data of 788 *Aeromonas* strains comprising 31 *Aeromonas* species, registered in the National Institutes of Health (NIH) genetic sequence database as of 20 June 2022, were downloaded (Table S2). The taxonomic affiliation of each genome was determined using ANI. These sequences were compared using the Python program for OSNAp (https://github.com/suzukimasahiro/OSNAp) in combination with BLAST to identify ORFs unique to each target species. ORFs specific to *A. hydrophila*, *A. caviae*, *A. veronii*, and *A. dhakensis* with a minimum sequence length of 200 bp were selected as candidate ORFs. Primers for each ORF were designed using Primer3Plus software (https://primer3plus.com) and the Oligo Evaluator web tool (http://www.oligoevaluator.com). Based on the multi-sequence alignment data generated by ClustalW in BioEdit software (https://bioedit.software.informer.com/), the primers were designed in highly conserved regions within each candidate ORF sequence. The sensitivity and specificity of each primer were initially evaluated by monoplex PCR using the clinical strains.

## Optimization of multiplex PCR

A primer mixture containing all primers was prepared, with the final concentration of each primer adjusted so that the band intensity of each gene fragment appeared approximately equal. A primer set targeting the 16S ribosomal RNA (rRNA) sequence was included in the multiplex PCR as an internal control, as described in a previous study (27). Multiplex PCR was carried out in a 20 µL final volume, containing 2.0 U of FastStart Taq DNA Polymerase (Roche, Basel, Switzerland), 2 µL of PCR buffer with $MgCl_2$ (10× concentration, containing 20 mM $MgCl_2$), 200 µM of each deoxynucleotide triphosphate (dNTP), primers at their adjusted concentrations, and 2 µL of template DNA. The thermal cycling conditions were as follows: pre-incubation at 95°C for 10 min, followed by 30 cycles of 95°C for 30 s, 60°C for 30 s, and 72°C for 1 min, with a final elongation at 72°C for 7 min. PCR amplicons were electrophoresed in 2.0% agarose gel for 45 min at 135 V and visualized by ethidium bromide staining and UV illumination.

## Evaluation of the accuracy of multiplex PCR

The accuracy of the multiplex PCR was assessed using 256 clinical *Aeromonas* strains and 19 type or reference strains. The multiplex PCR results for clinical strains were compared

with taxonomic affiliations obtained by WGS data, while for type or reference strains, they were compared with the registered genus–species names.

## RESULTS

Based on publicly available WGS data of *Aeromonas* strains, a total of 20 candidate ORFs were selected as species-specific markers for *A. hydrophila*, *A. caviae*, *A. veronii*, and *A. dhakensis,* which are listed in Table 1. Of these, five were for *A. hydrophila*, seven for *A. caviae*, four for *A. dhakensis*, and four for *A. veronii*. For each candidate ORF, a primer set was designed to yield a product of 100–400 bp. Initially, these primers were evaluated for their suitability as species markers through monoplex PCR on selected *Aeromonas* strains. Based on this evaluation, the final primer sets were chosen for the multiplex PCR. For *A. veronii,* for which no ORFs were consistently present across all strains, three primer

**TABLE 1** List of candidate ORFs for each target *Aeromonas* species

| Target species | Candidate ORFs | Location of candidate ORF in reference genomes (reference ACC no. _start_end) | Length (bp) | Evaluation by monoplex PCR | Suitability as a species-specific maker | Selected for mPCR |
|---|---|---|---|---|---|---|
| *A. caviae* | Acav_OFR_1 | LS483441.1_579939_580892 | 954 | Presence of extra bands in non-target species | NOT suitable | |
| | Acav_OFR_2 | LS483441.1_1288024_1288611 | 588 | Low positivity rate in target species | NOT suitable | |
| | Acav_OFR_3 | LS483441.1_2023827_2024913 | 1,101 | Presence of extra bands in non-target species | NOT suitable | |
| | Acav_OFR_4 | LS483441.1_3107519_3107986 | 471 | Presence of extra bands in non-target species | NOT suitable | |
| | Acav_OFR_5 | LS483441.1_30435_30953 | 519 | Presence of the target band only in target species | Suitable | |
| | Acav_OFR_6 | LS483441.1_3135133_3135645 | 513 | Presence of the target band only in target species | Suitable | Yes |
| | Acav_OFR_7 | LS483441.1_3437236_3438696 | 1,461 | Presence of extra bands in non-target species | NOT suitable | |
| *A. dhakensis* | Adha_ORF_1 | CP023141.1_2020192_2021601 | 1,410 | Presence of the target band only in target species | Suitable | Yes |
| | Adha_ORF_2 | CP023141.1_2018160_2019185 | 1,026 | Presence of the target band only in target species | Suitable | |
| | Adha_ORF_3 | CP023141.1_1540655_1541296 | 642 | NA[a] | NA | |
| | Adha_ORF_4 | CP023141.1_1595089_1595538 | 465 | NA | NA | |
| *A. hydrophila* | Ahyd_ORF_1 | NC_008570.1_2434963_2437512 | 2,553 | Presence of the target band in non-target species | NOT suitable | |
| | Ahyd_ORF_2 | NC_008570.1_2610907_2611287 | 381 | Presence of the target band only in target species | Suitable | Yes |
| | Ahyd_ORF_3 | NC_008570.1_3025437_3025649 | 213 | Presence of the target band in non-target species | NOT suitable | |
| | Ahyd_ORF_4 | NC_008570.1_3612337_3612585 | 249 | Low positivity rate in target species | NOT suitable | |
| | Ahyd_ORF_5 | NC_008570.1_3967841_3968305 | 465 | Presence of the target band in non-target species | NOT suitable | |
| *A. veronii* | Aver_ORF_1 | CP044060.1_3541178_3542008 | 831 | Presence of the target band only in target species | Suitable | Yes |
| | Aver_ORF_2 | CP044060.1_4313176_4314048 | 873 | Low positivity rate in target species | NOT suitable | |
| | Aver_ORF_3 | CP044060.1_2258308_2259051 | 744 | Presence of the target band only in target species | Suitable | Yes |
| | Aver_ORF_4 | CP044060.1_3100426_3101367 | 942 | Presence of the target band only in target species | Suitable | Yes |

[a]Abbreviation: NA, not available.

sets targeting distinct candidate ORFs were selected for the multiplex PCR (Table S3). The primer sequences used in the multiplex PCR, including those targeting 16S rRNA as an internal control, are listed in Table 2. The multiplex PCR was designed to produce the following target amplicon size: 124, 165, and 195 bp for *A. veronii*; 249 bp for *A. caviae*; 300 bp for *A. hydrophila*; 358 bp for *A. dhakensis*; and 461 bp for 16S rRNA. For *A. veronii*, detection of at least one of the three target fragments was considered positive. The final concentration of each primer used in the multiplex PCR is also listed in Table 2. As shown in Fig. 1A, amplicon patterns clearly distinguished each target species.

To validate this methodology, 256 clinical *Aeromonas* isolates and 11 *Aeromonas* type strains were subjected to multiplex PCR. Based on WGS data, the clinical strains were identified as follows: *A. caviae* (*n* = 117), *A. hydrophila* (*n* = 63), *A. veronii* (*n* = 44), *A. dhakensis* (*n* = 25), *A. allosaccharophila* (*n* = 2), *A. enteropelogenes* (*n* = 2), *A. media* (*n* = 1), *A. salmonicida* (*n* = 1), and *Aeromonas rivipollensis* (*n* = 1) (Table S4). As shown in Table 3, species identified by the multiplex PCR were consistent with those determined by WGS data or registered species names for all isolates belonging to the four target *Aeromonas* spp. For *A. veronii*, various combinations of amplicon patterns were observed: of the 45 strains, 21 (47%) showed three bands; 16 (38%) showed two bands; and 8 (18%) showed one band (Fig. 1B). No amplification products were obtained from isolates belonging to non-target *Aeromonas* spp., except for the 16S rRNA band. Furthermore, none of the type or reference strains from non-*Aeromonas* species (i.e., *E. coli*, *K. pneumoniae*, *E. cloacae* complex, *P. aeruginosa*, *V. fluvialis*, *V. parahaemolyticus*, *V. alginolyticus*, and *G. hollisae*) yielded positive bands with the multiplex PCR except for the 16S rRNA band. The gel image of multiplex PCR products from selected clinical strains is shown in Fig. S1.

## DISCUSSION

In this study, we developed and validated a one-step diagnostic multiplex PCR that identifies four *Aeromonas* spp., *A. hydrophila*, *A. caviae*, *A. veronii*, and *A. dhakensis*, which are frequently related to human infectious diseases.

Molecular methods have been the primary tools for taxonomic demarcation within the genus *Aeromonas* (2). Given the limited utility of the 16S rRNA gene due to high interspecies similarities (28), more advanced techniques have been implemented for species differentiation in this genus. Currently, multilocus phylogenetic analysis (MLPA) using several concatenated housekeeping genes (i.e., *gyrB*, *rpoD*, *recA*, *dnaJ*, *gyrA*, *dnaX*, and *atpD*) and WGS are regarded as the most accurate methods for identifying aeromonads to the species level (13, 29), although these techniques are time-consuming, labor-intensive, and costly. Among rapid species identification methods, the multiplex

**TABLE 2** Primers used in the multiplex PCR to identify the *Aeromonas* species

| Target organisms | Primers | Target ORFs (ACC no. _start_end) | Primer sequences (5′ to 3′, as synthesized) | Amplicon size (bp) | Primer concentration (µM) | Tm (°C) | GC (%) |
|---|---|---|---|---|---|---|---|
| *A. veronii* | Aver_ORF_4_F | CP044060.1_3100426_3101367 | GCAAGTGCAACTTCAAGCAA | 124 | 0.2 | 64 | 45 |
| | Aver_ORF_4_R | | TTTTTGGCATCCGTGGTATAG | | 0.2 | 63 | 43 |
| | Aver_ORF_1_F | CP044060.1_3541178_3542008 | CAACCTGTGCCAGAACTCCT | 165 | 0.2 | 64 | 55 |
| | Aver_ORF_1_R | | GGGTTCAAGCTTGTGAGGAC | | 0.2 | 64 | 55 |
| | Aver_ORF_3_F | CP044060.1_2258308_2259051 | CCATTGCTTGAGCATGAAGA | 195 | 0.2 | 64 | 45 |
| | Aver_ORF_3_R | | GTGCGAACCATCTACCAACC | | 0.2 | 64 | 55 |
| *A. caviae* | Acav_ORF_6_F | LS483441.1_3135133_3135645 | CCAGAGTCTCACATCCGTCA | 249 | 0.2 | 64 | 55 |
| | Acav_ORF_6_R | | ACGGCTCAACCAGGATCTC | | 0.2 | 64 | 58 |
| *A. hydrophila* | Ahyd_ORF_2_F | NC_008570.1_2610907_2611287 | CGCTCCTGCAATGCAATC | 300 | 0.4 | 66 | 56 |
| | Ahyd_ORF_2_R | | TTACCGCCGWGTGTTTGG | | 0.4 | 66 | 56 |
| *A. dhakensis* | Adha_ORF_1_F | CP023141.1_2020192_2021601 | CCCGATAACCAACCGTGAT | 358 | 0.2 | 65 | 53 |
| | Adha_ORF_1_R | | ATGCTGATCGGTGAAGGGTA | | 0.2 | 64 | 50 |
| | 16SrRNA_F (270) | (16S rRNA) | CGACGATCCCTAGCTGGTCT | 461 | 0.2 | 66 | 60 |
| | 16SrRNA_R (731) | | GCCTTCGCCACCGGTAT | | 0.2 | 66 | 65 |

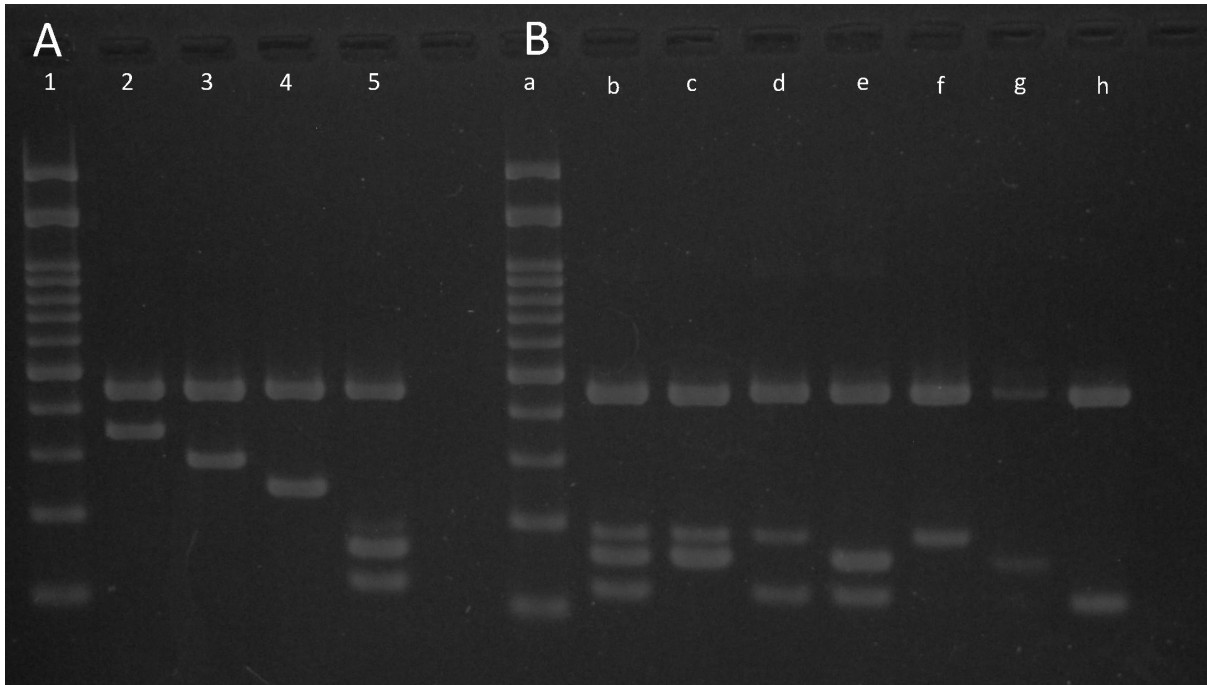

FIG 1 (A) Agarose gel electrophoresis patterns obtained by multiplex PCR for the identification of *Aeromonas* species. Six species-specific primer pairs were included in the multiplex PCR to produce unique amplicon sizes, along with a primer set targeting 16S rRNA as an internal control (461 bp). Lane 1, 100 bp DNA marker; Lane 2, *A. dhakensis* (clinical strain FUJ01730) (358 bp); Lane 3, *A. hydrophila* ATCC 7966[T] (300 bp); Lane 4, *A. caviae* JCM 1043[T] (249 bp); Lane 5, *A. veronii* JCM 7375[T] (124, 165, and 195 bp). (B) Multiplex PCR electrophoresis pattern of 11 clinical isolates of *A. veronii*. Lane a, 100 bp DNA marker; Lane b, *A. veronii* FUJH0261 (124, 165, and 195 bp); Lane c, *A. veronii* FUJH0479 (165 and 195 bp); Lane d, *A. veronii* FUJ00985 (124 and 195 bp); Lane e, *A. veronii* FUJ01278 (124 and 165 bp); Lane f, *A. veronii* FUJH1138 (195 bp); Lane g, *A. veronii* FUJ01909 (165 bp); Lane h, *A. veronii* FUJH0271 (124 bp).

PCR developed by Persson et al. differentiates four *Aeromonas* spp. (i.e., *A. hydrophila*, *A. media, A. caviae*, and *A. veronii*) using partial *gyrB* and *rpoB* gene sequences (27). However, this assay does not target *A. dhakensis*, an important pathogenic species reclassified from *A. hydrophila* subsp. *dhakensis* and *A. aquariorum* in 2013 and associated with high mortality in bacteremic patients (30, 31). Notably, *A. dhakensis* is the most prevalent species in warmer climate regions, such as Taiwan and Australia (32, 33). Our multiplex PCR assay, which includes *A. dhakensis* as a target species and is suitable for clinical use could be a valuable tool to support appropriate antimicrobial therapy and facilitate understanding of clinical characteristics, epidemiology, and pathogenicity of *Aeromonas* spp.

TABLE 3 Comparison of species identification based on whole genome sequencing data and multiplex PCR for 256 clinical *Aeromonas* strains, with registered taxonomic name compared with multiplex PCR for type strains

| Species identification based on whole genome sequencing data[a] | No. of isolates (no. of type strain) | Multiplex PCR identification | | | | |
|---|---|---|---|---|---|---|
| | | *A. caviae* | *A. hydrophila* | *A. veronii* | *A. dhakensis* | No. of amplicons except for 16S rRNA |
| *A. caviae* | 118 (1) | 118 | 0 | 0 | 0 | 0 |
| *A. hydrophila* | 64 (1) | 0 | 64 | 0 | 0 | 0 |
| *A. veronii* | 45 (1) | 0 | 0 | 45 | 0 | 0 |
| *A. dhakensis* | 25 (0) | 0 | 0 | 0 | 25 | 0 |
| Other *Aeromonas* spp.[b] | 15 (8) | 0 | 0 | 0 | 0 | 15 |
| Non-*Aeromonas* spp.[c] | 8 (8) | 0 | 0 | 0 | 0 | 8 |

[a]For type strains, registered genus–species names were used.
[b]Other *Aeromonas* spp. include *A. allosaccharophila* (n = 3), *A. enteropelogenes* (n = 3), *A. salmonicida* (n = 2), *A. media* (n = 2), *A. jandaei* (n = 1), *A. eucrenophila* (n = 1), *A. schubertii* (n = 1), *A. sobria* (n = 1), and *A. rivipollensis* (n = 1).
[c]Non-*Aeromonas* spp. include *Escherichia coli* (n = 1), *Enterobacter cloacae* subsp. *cloacae* (n = 1), *Klebsiella pneumoniae* (n = 1), *Pseudomonas aeruginosa* (n = 1), *Vibrio alginolyticus* (n = 1), *Vibrio fluvialis* (n = 1), *Vibrio parahaemolyticus* (n = 1), *Grimontia hollisae* (n = 1).

In the present study, ORF sequences specific to the target species were identified and optimized for multiplex PCR. This approach, where the distribution of ORFs in the study population is analyzed to detect sequences unique to specific species or genetic lineages, has been successfully applied in other taxonomic groups, such as *P. aeruginosa* (26), *Acinetobacter baumannii* (25) and *K. pneumoniae* (34). For *Aeromonas* spp., the advantage of using distinct ORFs as markers for each target species is that they are less affected by horizontal gene transfer, which frequently occurs in operational genes (e.g., housekeeping genes) within *Aeromonas* population, at rates as high as 5.8% (35). In fact, our ORF-based multiplex PCR assay demonstrated greater accuracy in identifying selected *Aeromonas* spp. compared to the aforementioned multiplex PCR assays based on housekeeping genes that reported 90.5% accuracy for *A. caviae* and 89% for *A. veronii*, according to a previous report (13, 27).

The strength of this study is that we used a large number of clinical *Aeromonas* strains to assess the efficacy of the multiplex PCR, with their taxonomic affiliations inferred from WGS data. On the other hand, several minor *Aeromonas* spp., which are rarely isolated from clinical specimens, were unavailable and could not be included in the evaluation of the multiplex PCR. Nevertheless, genome data of 31 *Aeromonas* spp. were analyzed *in silico* to identify unique ORF sequences for the target species and reduce the likelihood of non-specific amplification. Additionally, false-negative results could occur due to the absence of ORFs used as species-specific markers in this study, as demonstrated by *in silico* analysis of NCBI genomes. For example, *A. dhakensis* CP075589 in Table S3 was negative for all candidate ORFs tested.

In summary, we developed a one-step multiplex PCR assay to identify four clinically important *Aeromonas* spp., suitable for use in clinical microbiological laboratories. This PCR assay, combined with phenotypic antimicrobial susceptibility testing, has the potential to provide useful information to guide appropriate antimicrobial therapy and to enhance understanding of the epidemiology, clinical features, and pathogenicity of this complex taxon.

## ACKNOWLEDGMENTS

We thank the clinical laboratory staff for their assistance in collecting the bacterial strains used in the study.

The study was supported by the Japanese Association of Infectious Diseases funding for the promotion of clinical research. The collection of clinical strains was supported in part by Shionogi & Company, Limited.

## AUTHOR AFFILIATIONS

[1]Department of Infectious Diseases, Fujita Health University School of Medicine, Aichi, Japan

[2]Disease Control and Prevention Center, National Center for Global Health and Medicine, Tokyo, Japan

[3]Department of Infectious Diseases, Kameda Medical Center, Chiba, Japan

[4]Department of Infectious Diseases, Cancer Institute Hospital, Japanese Foundation for Cancer Research, Tokyo, Japan

[5]Department of Microbiology and Infectious Diseases, Toho University School of Medicine, Tokyo, Japan

[6]Department of Clinical Laboratory Medicine, Kyoto University Graduate School of Medicine, Kyoto, Japan

[7]Department of Infectious Diseases, Graduate School of Medical Sciences, Nagoya City University, Aichi, Japan

[8]Department of Infectious Diseases, St. Luke's International Hospital, Tokyo, Japan

[9]Department of Infectious Diseases, NTT Medical Center, Tokyo, Japan

[10]Department of Infectious Diseases, Japanese Red Cross Narita Hospital, Chiba, Japan

[11]Hospital Infection Control Center, Oita University Hospital, Oita, Japan

¹²Department of Microbiology, Fujita Health University School of Medicine, Aichi, Japan
¹³Center for Infectious Disease Research, Fujita Health University, Aichi, Japan
¹⁴Center for Innovative Antimicrobial Therapy, Division of Infectious Diseases, University of Pittsburgh School of Medicine, Pittsburgh, Pennsylvania, USA

**AUTHOR ORCIDs**

Aki Sakurai 🔟 http://orcid.org/0000-0003-2198-8954
Sohei Harada 🔟 http://orcid.org/0000-0003-3073-6564
Yasufumi Matsumura 🔟 http://orcid.org/0000-0001-8595-8944
Yohei Doi 🔟 http://orcid.org/0000-0002-9620-2525
Masahiro Suzuki 🔟 http://orcid.org/0000-0003-4550-3499

**FUNDING**

| Funder | Grant(s) | Author(s) |
| --- | --- | --- |
| The Japanese Association of Infectious Diseases | | Aki Sakurai |
| Shionogi & Co. Ltd. | | Sho Saito |

**AUTHOR CONTRIBUTIONS**

Aki Sakurai, Conceptualization, Data curation, Formal analysis, Funding acquisition, Investigation, Methodology, Resources, Visualization, Writing – original draft | Naoto Hosokawa, Investigation, Resources, Writing – review and editing | Daisuke Ohkushi, Investigation, Resources, Writing – review and editing | Sohei Harada, Investigation, Resources, Writing – review and editing | Yasufumi Matsumura, Investigation, Resources, Writing – review and editing | Naoya Itoh, Investigation, Resources, Writing – review and editing | Kazuhiro Ishikawa, Investigation, Resources, Writing – review and editing | Sho Saito, Data curation, Resources, Writing – review and editing | Takayuki Sakurai, Investigation, Resources, Writing – review and editing | Ryota Hase, Investigation, Resources, Writing – review and editing | Takehiro Hashimoto, Investigation, Resources, Writing – review and editing | Yohei Doi, Funding acquisition, Supervision, Writing – review and editing | Masahiro Suzuki, Conceptualization, Investigation, Software, Supervision, Validation, Writing – review and editing

**DATA AVAILABILITY**

The genome sequencing data of clinical *Aeromonas* strains were deposited under BioProject accession numbers PRJNA896347 and PRJNA1040422.

**ADDITIONAL FILES**

The following material is available online.

Supplemental Material

**Fig. S1 (Spectrum03331-24-s0001.tif).** Gel image of multiplex PCR products from selected clinical strains.
**Supplemental legends (Spectrum03331-24-s0002.docx).** Legends for supplemental figures and tables.
**Table S1 (Spectrum03331-24-s0003.docx).** Reference genomes representing 31 *Aeromonas* species.
**Table S2 (Spectrum03331-24-s0004.xlsx).** List of NCBI *Aeromonas* genomes.
**Table S3 (Spectrum03331-24-s0005.xlsx).** Presence or absence of candidate ORFs in NCBI *Aeromonas* genomes.
**Table S4 (Spectrum03331-24-s0006.xlsx).** List of clinical *Aeromonas* strains.

## Open Peer Review

**PEER REVIEW HISTORY (review-history.pdf).** An accounting of the reviewer comments and feedback.

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
