## [Reviewer comments · Microbiology Spectrum]

Microbiology Spectrum

Multiplex PCR assay to identify clinically important *Aeromonas* species

Aki Sakurai, Naoto Hosokawa, Daisuke Ohkushi, Sohei Harada, Yasufumi Matsumura, Naoya Itoh, Kazuhiro Ishikawa, Sho Saito, Takayuki Sakurai, Ryota Hase, Takehiro Hashimoto, Yohei Doi, and Masahiro Suzuki

Corresponding Author(s): Yohei Doi, University of Pittsburgh School of Medicine

Review Timeline:

Submission Date:	December 19, 2024
Editorial Decision:	January 18, 2025
Revision Received:	February 6, 2025
Accepted:	March 5, 2025

Editor: Po-Yu Liu

Reviewer(s): Disclosure of reviewer identity is with reference to reviewer comments included in decision letter(s). The following individuals involved in review of your submission have agreed to reveal their identity: Michael Wehrhahn (Reviewer #1)

Transaction Report:

DOI: <https://doi.org/10.1128/spectrum.03331-24>

Re: Spectrum03331-24 (Multiplex PCR assay to identify clinically important *Aeromonas* species)

Dear Dr. Yohei Doi:

Thank you for the privilege of reviewing your work. Below you will find my comments, instructions from the Spectrum editorial office, and the reviewer comments.

Revision Guidelines

Sincerely,
Po-Yu Liu
Editor
Microbiology Spectrum

Reviewer #1 (Comments for the Author):

This is an important study containing useful validation of a multiplex PCR covering the 4 most common *Aeromonas* spp which cause the majority of human infections. Using distinct ORFs as targets, they improve the accuracy of previously published multiplex PCRs and include the important *A. dhakensis* species.

Only minor suggestions are provided below:

61 could strengthen this and say ~90-98% if include ref 5

88 - suggest listing a few other sites making up remainder or at least the proportion wound/abscess or skin/soft tissue infection

samples

155 Supp Table 3: A ver ORF 1,2,3,4: misses 23,10,15,37 of infections respectively whereas Table 1 indicates that ORF 2 missed the most and was not suitable which seems the opposite and appeared to be best in this Supp Table

160 Table 1 > Table 2

170 ampliation > amplification products/amplicons

182 multi-locus sequence typing (MLST) rather than MLSA I think is being described here

192 suggest could add sentence elaborating for example *A. caviae* likely to be S to meropenem while *A. dhakensis* better treated with cefepime due to higher proportion with metallo beta-lactamase (ref 5) or other such clinical example to further highlight importance of correct speciation of this genus

312 Table 1: start-end appear to be reversed for *A. dhakensis* and ORF 3 and 4 for *A. veronii*

319 *A. enteropelogenes* > *A. enteropelogenes*

Reviewer #2 (Comments for the Author):

Sakurai et al. developed a multiplex PCR assay to identify clinically important *Aeromonas* species. The manuscript is interesting and well written, it highlights the benefits of rapid *Aeromonas* species identification, and it includes a high number of clinically important *Aeromonas* species to validate the PCR.

Minor comments:

Lines 73-75: Another limitation of MALDI-TOF for species identification of *Aeromonas* that should be mentioned is the incomplete databases, which in some cases contain a single representative or a reduced number of strains of each species.

Lines 200-202: While the PCR developed for this study showed 100% accuracy for the isolates tested, the authors should mention that since it is possible that some isolates may miss the ORFs used for identification (as shown by *A. dhakensis* CP075589 in table S3, which was negative for all 4 ORFs tested), false negative results could still potentially occur.

Lines 210-212: While the PCR can be useful to guide treatment, it should be noted that it is not meant to replace phenotypic antimicrobial susceptibility testing.

Reviewer #3 (Comments for the Author):

The Manuscript by Aki Sakurai and colleagues describes the setup of one-step multiplex PCR for the diagnostic identification of four *Aeromonas* spp. involved in human infections. The Authors considered the ORF regions, inferring data from published WGS sequences. The study's strength is the validation of the multiplex PCR in many *Aeromonas* clinical isolates with no signals detected in non-*Aeromonas* standard strains. As compared to already published studies, this one also includes *A. dhakensis*.

Main considerations:

- In Figure 1B, the Authors reported data about *A. veronii* clinical isolates. Since one of the main points of the Manuscript is its inclusion in the multidetection system of *A. dhakensis*, gel images about this species should be included.
- In the Introduction, the Authors focus on the importance of antibiotic resistance in *Aeromonas* spp. However, the main goal of the Manuscript is not the detection of resistance genes. Indeed, I suggest including more information about the clinical importance, epidemiology, and virulence factors of *Aeromonas* spp infections in humans, taking into consideration recently published papers, such as doi: 10.1128/spectrum.03705-22; doi: 10.1128/spectrum.00807-23; doi: 10.1093/cid/ciae272.
- Keywords should be enriched with others, more informative ones, such as human infection, species identification, rapid identification, and so on.

Multiplex PCR assay to identify clinically important *Aeromonas* species by Sakurai et al

This is an important study containing useful validation of a multiplex PCR covering the 4 most common *Aeromonas* spp which cause the majority of human infections. Using distinct ORFs as targets, they improve the accuracy of previously published multiplex PCRs and include the important *A. dhakensis* species.

Only minor suggestions are provided below:

61 could strengthen this and say ~90-98% if include ref 5

88 – suggest listing a few other sites making up remainder or at least the proportion wound/abscess or skin/soft tissue infection samples

155 Supp Table 3: *A. veronii* ORF 1,2,3,4: misses 23,10,15,37 of infections respectively whereas Table 1 indicates that ORF 2 missed the most and was not suitable which seems the opposite and appeared to be best in this Supp Table

160 Table 1 > Table 2

170 ampliation > amplification products/amplicons

182 multi-locus sequence typing (MLST) rather than MLSA I think is being described here

192 suggest could add sentence elaborating for example *A. caviae* likely to be S to meropenem while *A. dhakensis* better treated with cefepime due to higher proportion with metallo beta-lactamase (ref 5) or other such clinical example to further highlight importance of correct speciation of this genus

312 Table 1: start-end appear to be reversed for *A. dhakensis* and ORF 3 and 4 for *A. veronii*

319 *A. enteropelogenes* > *A. enteropelogenes*

The Manuscript by Aki Sakurai and colleagues describes the setup of one-step multiplex PCR for the diagnostic identification of four *Aeromonas* spp. involved in human infections. The Authors considered the ORF regions, inferring data from published WGS sequences. The study's strength is the validation of the multiplex PCR in many *Aeromonas* clinical isolates with no signals detected in non-*Aeromonas* standard strains. As compared to already published studies, this one also includes *A. dhakensis*.

Main considerations:

- In Figure 1B, the Authors reported data about *A. veronii* clinical isolates. Since one of the main points of the Manuscript is its inclusion in the multidetection system of *A. dhakensis*, gel images about this species should be included.

- In the Introduction, the Authors focus on the importance of antibiotic resistance in *Aeromonas* spp. However, the main goal of the Manuscript is not the detection of resistance genes. Indeed, I suggest including more information about the clinical importance, epidemiology, and virulence factors of *Aeromonas* spp infections in humans, taking into consideration recently published papers, such as doi: 10.1128/spectrum.03705-22; doi: 10.1128/spectrum.00807-23; doi: 10.1093/cid/ciae272.

- Keywords should be enriched with others, more informative ones, such as human infection, species identification, rapid identification, and so on.

Dear Editors and Reviewers

We wish to express our appreciation to the Editors and Reviewers for the insightful comments, which have helped us significantly improve the paper. We have addressed these comments with point-by-point responses, and revised the manuscript accordingly.

Response to Reviewer 1:

Reviewer #1 (Comments for the Author):

This is an important study containing useful validation of a multiplex PCR covering the 4 most common *Aeromonas* spp which cause the majority of human infections. Using distinct ORFs as targets, they improve the accuracy of previously published multiplex PCRs and include the important *A. dhakensis* species.

Only minor suggestions are provided below:

61 could strengthen this and say ~90-98% if include ref 5

Response: The suggested change has been made, with reference 5 included.

(Line 62)

88 - suggest listing a few other sites making up remainder or at least the proportion wound/abscess or skin/soft tissue infection samples

Response: Per the Reviewer's suggestion, the proportions of wound and intra-abdominal specimens have been added to the revised manuscript. (Line 102-103)

155 Supp Table 3: *A. veronii* ORF 1,2,3,4: misses 23,10,15,37 of infections respectively whereas Table 1 indicates that ORF 2 missed the most and was not suitable which seems the opposite and appeared to be best in this Supp Table

Response: We thank the Reviewer for the insightful comments. In Supp Table 3, candidate ORF regions with $\geq 80\%$ nucleotide sequence identity and $\geq 80\%$ coverage were considered "present" in each genome and displayed as colored blocks accordingly. As the Reviewer pointed out, although *A. veronii* ORF2 (length 873bp) was present in most isolates, the genomic regions suitable for primer design were limited. Consequently, primers were designed in regions where several strains contained point mutations, which likely

contributed to the low positivity rate in the monoplex PCR. To improve clarity, we added further explanations (e.g., the definition of “the presence of candidate ORF” in each genome) in Supp Table 3. (Supplementary Appendix)

160 Table 1 > Table 2

Response: We have corrected it accordingly. (Line 174)

170 ampliation > amplification products/amplicons

Response: We have corrected it accordingly. (Line 184)

182 multi-locus sequence typing (MLST) rather than MLSA I think is being described here

Response: As suggested, we changed multilocus sequence analysis (MLSA) to multilocus phylogenetic analysis (MLPA) in accordance with reference 13. (Line 197)

192 suggest could add sentence elaborating for example *A. caviae* likely to be S to meropenem while *A. dhakensis* better treated with cefepime due to higher proportion with metallo beta-lactamase (ref 5) or other such clinical example to further highlight importance of correct speciation of this genus

Response: Per the Reviewer’s suggestion, we have added a sentence explaining the importance of correct speciation in selecting appropriate antimicrobial agents, including an example. (Line 69-73)

12 Table 1: start-end appear to be reversed for *A. dhakensis* and ORF 3 and 4 for *A. veronii*

Response: We have corrected them. (Table 1, 2) (Line 366-368)

319 *A. enteropelogenes* > *A. enteropelogenes*

Response: We have corrected it accordingly. (Line 373)

Response to Reviewer 2:

Reviewer #2 (Comments for the Author):

Sakurai et al. developed a multiplex PCR assay to identify clinically important *Aeromonas* species. The manuscript is interesting and well written, it highlights the benefits of rapid *Aeromonas* species identification, and it includes a high number of clinically important *Aeromonas* species to validate the PCR.

Minor comments:

Lines 73-75: Another limitation of MALDI-TOF for species identification of *Aeromonas* that should be mentioned is the incomplete databases, which in some cases contain a single representative or a reduced number of strains of each species.

Response: We thank the Reviewer for the insightful comments. Per the Reviewer's suggestion, we have addressed the limitation of MALDI-TOF related to incomplete databases in the revised manuscript. (Line 80-82)

Lines 200-202: While the PCR developed for this study showed 100% accuracy for the isolates tested, the authors should mention that since it is possible that some isolates may miss the ORFs used for identification (as shown by *A. dhakensis* CP075589 in table S3, which was negative for all 4 ORFs tested), false negative results could still potentially occur.

Response: We agree with the Reviewer's comment. We have addressed it in the Discussion section. (Line 224-227)

Lines 210-212: While the PCR can be useful to guide treatment, it should be noted that it is not meant to replace phenotypic antimicrobial susceptibility testing.

Response: We have revised the manuscript in accordance with the Reviewer's suggestion.

(Line 230)

Response to Reviewer 3:

Reviewer #3 (Comments for the Author):

The Manuscript by Aki Sakurai and colleagues describes the setup of one-step multiplex PCR for the diagnostic identification of four *Aeromonas* spp. involved in human infections. The Authors considered the ORF regions, inferring data from published WGS sequences. The study's strength is the validation of the multiplex PCR in many *Aeromonas* clinical isolates with no signals detected in non-*Aeromonas* standard strains. As compared to already published studies, this one also includes *A. dhakensis*.

Main considerations:

- In Figure 1B, the Authors reported data about *A. veronii* clinical isolates. Since one of the main points of the Manuscript is its inclusion in the multidetection system of *A. dhakensis*, gel images about this species should be included.

Response: We appreciate the valuable comments from the Reviewer. Per the Reviewer's suggestion, the gel image of clinical strains, including those of *A. dhakensis*, is presented in Supplementary Fig 1. (Line 188-189)

- In the Introduction, the Authors focus on the importance of antibiotic resistance in *Aeromonas* spp. However, the main goal of the Manuscript is not the detection of resistance genes. Indeed, I suggest including more information about the clinical importance, epidemiology, and virulence factors of *Aeromonas* spp infections in humans, taking into consideration recently published papers, such as doi: 10.1128/spectrum.03705-22; doi: 10.1128/spectrum.00807-23; doi: 10.1093/cid/ciae272.

Response: We thank the Reviewer for the helpful comment. We have now included additional information on the clinical importance of accurate speciation, which is essential for understanding the clinical features, epidemiology, and pathogenicity of human *Aeromonas* infections, as supported by recent studies. (Line 83-89)

- Keywords should be enriched with others, more informative ones, such as human infection, species identification, rapid identification, and so on.

Response: The suggested change has been made. (Line 27-28)

Re: Spectrum03331-24R1 (Multiplex PCR assay to identify clinically important *Aeromonas* species)

Dear Dr. Yohei Doi:

Your manuscript has been accepted, and I am forwarding it to the ASM production staff for publication. Your paper will first be checked to make sure all elements meet the technical requirements. ASM staff will contact you if anything needs to be revised before copyediting and production can begin. Otherwise, you will be notified when your proofs are ready to be viewed.

Sincerely,
Po-Yu Liu
Editor
Microbiology Spectrum

Reviewer #1 (Comments for the Author):

COmments have been adequately addressed

Reviewer #2 (Comments for the Author):

The authors have appropriately addressed all my comments in the response to the reviewers.

Reviewer #3 (Comments for the Author):

none